# Hormones and Sex-Specific Medicine in Human Physiopathology

**DOI:** 10.3390/biom12030413

**Published:** 2022-03-07

**Authors:** Maria Raza Tokatli, Leuconoe Grazia Sisti, Eleonora Marziali, Lorenza Nachira, Maria Francesca Rossi, Carlotta Amantea, Umberto Moscato, Walter Malorni

**Affiliations:** 1Course in Pharmacy, University of Tor Vergata, 00133 Rome, Italy; mariatokatli3@gmail.com; 2Center for Global Health Research and Studies, Università Cattolica del Sacro Cuore, 00168 Rome, Italy; leuconoegrazia.sisti@unicatt.it (L.G.S.); eleonora.marziali01@icatt.it (E.M.); lorenza.nachira01@icatt.it (L.N.); umberto.moscato@unicatt.it (U.M.); 3National Institute for Health, Migration and Poverty, 00153 Rome, Italy; 4Department of Life Sciences and Public Health, Section of Occupational Health, Università Cattolica del Sacro Cuore, 00168 Rome, Italy; mariafrancesca.rossi01@icat.it (M.F.R.); carlotta.amantea@unicatt.it (C.A.)

**Keywords:** gender medicine, sex hormones, neurodegenerative diseases, neurological diseases, cancer, immunity, infectious diseases, cardiovascular diseases

## Abstract

A prodigious increment of scientific evidence in both preclinical and clinical studies is narrowing a major gap in knowledge regarding sex-specific biological responses observed in numerous branches of clinical practices. Some paradigmatic examples include neurodegenerative and mental disorders, immune-related disorders such as pathogenic infections and autoimmune diseases, oncologic conditions, and cardiovascular morbidities. The male-to-female proportion in a population is expressed as sex ratio and varies eminently with respect to the pathophysiology, natural history, incidence, prevalence, and mortality rates. The factors that determine this scenario incorporate both sex-associated biological differences and gender-dependent sociocultural issues. A broad narrative review focused on the current knowledge about the role of hormone regulation in gender medicine and gender peculiarities across key clinical areas is provided. Sex differences in immune response, cardiovascular diseases, neurological disorders, cancer, and COVID-19 are some of the hints reported. Moreover, gender implications in occupational health and health policy are offered to support the need for more personalized clinical medicine and public health approaches to achieve an ameliorated quality of life of patients and better outcomes in population health.

## 1. Introduction

Deliberate contemplation of sex-biased data in preclinical and clinical studies was initiated only in the last few decades, howbeit a historical assertion of sex differences in the manifestation of diseases and response to treatment was already stated by Hippocrates more than 2000 years ago. Contemporary scientific research endeavors to consolidate sex and gender as key components in clinical trials, allowing more realistic evaluations. Race and cultural identities and psychosocial and economic statuses, however, are all restricting factors that conduce to a certain incongruity, predominantly with women being much less enrolled in experimental practices. Designing clinical trials with an equally adjusted sex–gender recruitment is a rather challenging task, especially due to reproductive age, but also pregnancy and lactating periods during a woman’s life, which greatly limit their participation in drug trials. Conversely, in certain circumstances (e.g., in studies carried out on neurological diseases or on autoimmune diseases), due to their predominance, the number of women is significantly higher, and this could represent a critical issue. A further point to be considered concerns the important diurnal fluctuations of hormones (e.g., testosterone), which receive limited consideration in experimental studies that do not involve hormonal therapy. This could represent a further bias altering the clinical image and therapeutic decisions.

First, what is important to highlight is the divergent terminology of sex and gender which, conventionally, are mistakenly used as synonyms. Sex refers to the biological and genetic features of individuals, whereas gender is intended as the social perspective of human beings concerning expressions, behaviors, and social roles. The latter is considered a personal choice and can generate a nonbinary range of gender identities. While some inter-relationship between the hormonal and genetic milieu and gender identity selection may exist, the exact mechanisms are still unknown. This phenomenon is rather troublesome to investigate due to the fact that humans present consciousness, a trait that is still at least unclear in animals, and research in this area is limited [1].

Sex-specific biological differences are currently recorded in numerous branches of clinical practices influencing the physiology, pathophysiology, clinical manifestation, natural history, incidence, prevalence, treatment response, and mortality rates of key diseases. Similarly, gender-dependent sociocultural issues also affect the epidemiology and the diseases’ course, such as in the case of risky lifestyle attitudes or access to healthcare. This narrative review deals with biological substrata of gender differences in health and gender peculiarities across key clinical areas. Furthermore, an overview of gender implications in occupational health and health policymaking is also provided.

## 2. Sex Differences in Physiology: The Role of Sex Steroid Hormones

### 2.1. Brief Overview on the Synthesis and Roles of Sex Hormone Steroids

Sex steroid hormones are responsible for behavior and sexual differentiation, but also hold a key role in the normal function of various organs, including the brain. The major sex steroid hormones include estrogens, specifically estrone (E1), estradiol (E2), and estriol (E3); progestogen, including progesterone; and androgens, with the most important one being testosterone. The latter is responsible for the regulation of the reproductive organs of men, as well as the masculine characteristics, including the growth of bone mass and body hair. Testosterone is mainly produced by the adrenal gland and male gonad glands, the testes, from its precursor, cholesterol [2], and changes significantly with age. The release of sex hormones is regulated by the hypothalamic–pituitary–gonadal system; peripheral sex hormones provoke the discharge of gonadotropin-releasing hormone (GnRH) in the hypothalamus. GnRH generates the secretion of luteinizing hormone (LH) and follicle-stimulating hormone (FSH) in the pituitary that stimulate the release of sex hormones from the reproductive organs [3]. In particular, testicular steroidogenesis occurs in the Leydig cells, where the LH hormone stimulates enzymes essential for the sex hormone biosynthesis [4]. Specifically, the steroidogenic acute regulatory protein (StAR) assists cholesterol inside the mitochondria, where its side chain is cleaved by the P450scc enzyme and gets converted into pregnenolone. The P450c17 enzyme with its dual regulatory action mediates the consecutive steps, hydroxylating pregnenolone into 17-OH pregnenolone by 17α-hydroxylase, and further converting it into dehydroepiandrosterone (DHEA) by 17/20 lyase. The 17α-hydroxylase enzyme is also responsible for the conversion of progesterone to 17-OH progesterone, while 17/20 lyase further converts it into androstenedione. Additional enzymes, including 3β and 17β hydroxysteroid dehydrogenase, are essential for the DHEA conversion into androstenedione and its subsequent metabolism to testosterone, respectively. Testosterone is finally converted by 5α-reductase into dihydrotestosterone and 17β-estradiol by aromatase. The aromatization process of testosterone does occur in males on a small scale and mostly peripherally [5].

Contrarily, in females, the ovarian thecal cells along with the LH hormone are responsible for the initiation of the sex hormone synthesis. An upregulation of the aromatization reaction due to greater levels of LH hormones during the follicular phase of the estrous cycle results in elevated levels of estrogens, especially E2. In the subsequent luteal phase, a decrease in LH hormone and aromatization result in a decline in estrogen levels and an increase in progesterone [4]. Hence, female sex hormones present a considerable fluctuation pattern during the menstrual cycle, a feature that has hindered women’s enrolment in clinical studies but should undoubtedly be considered as it is responsible for considerable sex differences in the pathophysiology and treatment of a myriad of health conditions. Several studies have been executed in order to estimate the exact phases of the menstrual and estrous cycle and the related serum hormone concentrations [6]. Likewise, a fluctuation pattern of testosterone has been detected in males. These levels seem proportional to sunlight and sleep duration; peak levels have been observed during diurnal measurements, which reach a plateau and begin to drop in the afternoon. The decline seems to be shortened with advancing age. Restlessness has also been associated with decreased levels of testosterone [7,8].

Women also produce a small amount of testosterone by the adrenal glands and by the ovaries. The female sex hormones estrogens and progesterone are critical for determining women’s secondary characteristics, including breast growth and the body’s preparation for pregnancy, respectively [2]. Additionally, sex hormones are largely present on different sites of the brain, where they facilitate the regular neurological function and synaptic plasticity. They are not only produced peripherally by the reproductive organs. A plethora of studies have determined that approximately 2% of free plasma gonadal hormones synthesized peripherally cross the blood–brain barrier, and others are directly developed locally, de novo, from cholesterol-derived precursors [9]. The steroidogenesis in the brain gives rise to neurosteroids, which may function as inhibitory or excitatory neuromodulators upon direct interaction with steroid neuroreceptors or by allosteric regulation of ion channels [10]. Allopregnanolone, androstanediol, and tetrahydrodeoxycorticosterone (THDOC) are among the most important neurosteroids [11].

As mentioned above, sexual differentiation is also ascribed to the early production of sex hormones. In detail, whereas the genetic sex of an embryo is acquired at fertilization by inhering either an XY pair of chromosomes for males or XX for females, sexual differentiation involves the progressive development of the gonads and genitalia towards a male or female profile. In the first case, the release of anti-Müllerian hormone (AMH) and androgens—specifically testosterone produced by the Leydig cells—promotes the development of male internal and external genitalia, suppressing the female morphology. On the contrary, the female pathway is promoted in the absence of the male hormones. While in the past it was believed that solely the SRY gene present on the Y chromosome was responsible for gonadal differentiation, recent studies suggest that a wider group of genes is in play [1,12]. Thus, the release or absence of testicular sex hormones is also crucial for brain development—occurring during the second half of pregnancy—and sexual behavior by masculinization or feminization in the brain, respectively. However, as observed in studies of mammals and birds, specific expression or repression of genes in sex chromosomes has been found to alter the route of brain sexual differentiation regardless of the gonadal hormones [13,14]. Since brain development is not simultaneous to genital differentiation, the gender identity and sexual orientation of an individual might not be consistent with his or her chromosomal pair. Further investigation on brain early exposure to sex hormones prenatally or exposure during pregnancy to endocrinological modulators, including exogenous endocrine-disrupting agents, is essential to better comprehend their potential impact on gender diversity [15].

### 2.2. On the Role of Sex Steroid Hormones and Their Receptors

Sex hormones interact with specific receptors located in the cytoplasm or nucleus of target cells or at the cell membrane [9]. Identified estrogen receptors consist of nuclear estrogen receptors (ERs) and non-nuclear estrogen receptors, both including estrogen receptor alpha (ERα) and estrogen receptor beta (ERβ), and G protein-coupled estrogen receptors (GPERs), membrane receptors that are also encountered in the endoplasmic reticulum of many cell types [16]. More precisely, the activation of the receptors starts when estrogens bind to cytoplasmic ERs, which are then translocated to the nucleus. The ligand–receptor complex binds to a hormone response element, a particular supervisory DNA segment in the promoter region of genes, leading to the modulation of transcription and the mRNA modification [17]. More recent literature suggests that, apart from the typical genomic interaction of estrogens with their receptors, the binding of non-nuclear ERs may trigger prompt signaling pathways involved in various cell functions and has been detected in different histotypes, including nervous system cells and lymphocytes [9,17,18,19]. Classical, nuclear acting ERs can function as transcription factors and regulate gene expression on a slow time range. These moderately long processes had been puzzling scientists as they could not explain the observed rapid changes in neurological or immune system behavior, alluding to the presence of another type of ERs. This hypothesis was initially experimented by investigators who realized that estradiol provision in the tissue of the uterine cavity resulted in rapid increment of cAMP, which was due to estradiol binding to cell membrane hormone receptors. This concept was later corroborated by additional analysis, which has also observed the rapid increased phosphorylation of the transcription factor cAMP-responsive element-binding protein (CREB) upon elevated estradiol concentrations mediated via the MAPK/ERK system [20,21].

A supplementary, still uncharted function of ERs is related to their expression on the mitochondrion. Preclinical studies have deduced the presence of ER–estrogen complexes in the mitochondria of primary neuronal cultures, as well as in the mitochondria of hippocampal presynaptic and postsynaptic neurons of rodents. However, data are still restricted, and further exploration of the exact mechanisms is necessary. This regulation of the mitochondria, however, along with potential mtDNA mutations that may accumulate as time elapses, is believed to be associated with brain aging and other neurodegenerative disorders [22].

## 3. Sex Differences in Pathology

### 3.1. Sex Differences in Neurologic Diseases

As mentioned above, sex hormones play a role in several neurological and psychiatric diseases. This role is essentially due to their regulatory activity in neuronal signal conduction and, thus, brain functioning. Nevertheless, due to the difficulties in including sex hormones as a biological variable while investigating new approaches in the comprehension and treatment of disorders, their role is often neglected. Recent animal research studies have attempted to clarify the sex disparities observed in neurological disorders, but they might not always be consistent with human studies, further complicating the scene. For example, in the case of male rodents, in which aromatase transforms testosterone to estradiol in the developing fetal brain, excess estradiol levels are prevented by alpha-fetoprotein’s restraining competence. On the contrary, in human populations, the sex hormone-binding globulin (SHBG) protein has been found to have analogous action to alpha-fetoprotein but presents a higher sensitivity for androgens [23]. Overall, the existing knowledge underlines the sex hormones’ influence on some of the most paradigmatic neurological disorders, such as Alzheimer’s disease, Parkinson’s disease, epilepsy, but also depression and schizophrenia.

Alzheimer’s disease. Alzheimer’s disease (AD) is a progressive neurodegenerative disorder of the brain that includes a broad range of manifestations, including confusion, language difficulties, visuospatial dysgnosia, memory decline, thinking impairment, and a wide spectrum of different severities, from a mild cognitive impairment to the development of frank dementia. AD may impact both male and female populations; however, a marked dimorphism has been observed. Women present a higher prevalence of AD onset and progression. Women’s longevity may partially explain this predominance, but reflections on sex hormones should be embodied as probable risk factors along with age. As the aging process occurs, women are exposed to a higher risk of suffering from the disease with a 0.7% frequency of occurrence in ages 65–69 years, while men’s risk is 0.6% in the same age group. Interestingly, this probability is amplified in ages 86–89 years to 14.2% in women in comparison with 8.8% in men [24]. Multiple studies in Europe have revealed a higher probability of diagnosis of dementia in female AD patients, results that do not correspond to studies carried out in the U.S. or other countries. This disparity might indicate that, beyond sex, the geographical area or even a limited educational access may also be considered a risk factor for AD dementia. Thus, several other elements that may have an impact on the living routine of individuals shall not be excluded. For instance, depression, which is detected more frequently in women, or sleep apnea, which predominates in men and menopausal women, result in disturbances in human habits throughout life that might be associated with impaired awareness or cognitive deficits. Overall, women also exhibit a higher rate of different forms of cognitive deterioration [25,26]. Specific studies concerning AD and dementia’s neuropsychiatric symptoms affecting behavior have shown a comparatively higher likelihood of women to develop depression and anxiety, while agitation and hostile behavior have been more frequently reported in men [26].

Apart from sociopsychological risk factors, biological features, such as brain structure, sex hormones, and genotypic differences, are indubitably involved. Several studies have been conducted comparing the female and male brain in order to verify or refute myths involved in neurosexism [27]. The truth is that there are some differences in brain structure and function between the sexes regardless of data from previous studies being divergent. It has been surmised that the male brain is generally larger in volume and that females have a proportionally larger amount of grey matter in comparison with white matter in different compartments of the brain [28]. Differences in brain function may also indicate a contribution to a decline in cognition, as females acquire a higher activity in the sensory association cortex of the parietal lobe, while males in the motor and visual cortices [29]. Although according to the present literature the presence of Amyloid-β (Aβ) plaques, an AD biomarker, does not show any clear sex differences in the distribution in brain areas, a correlation of plaques with the rate of neurodegeneration has been observed. More specifically, the female brain exhibits more atrophic hippocampal conditions with a quicker cognitive impairment than the male one, even in patients with mild cognitive impairment (MCI). In this regard, the levels of another AD biomarker, the tau tangle formation, have also been studied, reporting sometimes controversial results. However, a higher tau aggregation in specific brain cortical segments of AD women and a general elevated concentration of tau in MCI patients have been observed [30]. Further studies analyzing the toxicity of Aβ aggregates highlight the role covered by oxidative damage. It has been reported that Aβ peptides are capable of binding to the heme group of the substrates participating in the respiratory chain of the mitochondrion and lead to an increased release of reactive oxygen species. Acknowledging this mechanism, the protective antioxidant properties of female sex hormones in young adults, which decline with age, have also been called into play [31].

Gonadal sex hormones’ role in preserving memory and cognitive deficits through the dopaminergic system and by diminishing Aβ accumulation in AD has now been uncovered. In women, the rapid decline of estrogens during menopause contributes to the AD prevalence. Autopsy analyses in AD patients have shown reduced estrogen and androgen levels in the brain in women and men, respectively, suggesting a modification of the brain hormonal synthesis in AD patients [32]. In fact, lower levels of free estradiol and higher levels of SHBG have been affiliated to a more rapidly developing cognitive decline, although contradicting results from studies also exist [33].

In this regard, data have proposed hormonal therapy as a cognitive impairment protective strategy in women only when started during perimenopause or in close temporal proximity to a sudden change in female hormonal levels (e.g., in the case of bilateral oophorectomy) [34].

In addition, sex hormones are known to influence genes that have been suggested to be predisposing factors for AD, for instance, the ApoE gene alleles. Studies have demonstrated that women who are carriers of the ApoE4 allele have a drastically more rapid cognitive decline compared with male patients with similar genotypes, but also present an overexpression of the β-site APP-cleaving enzyme (BACE1), which is crucial for Aβ production. A plausible explanation might be searched in the modulatory role of ERs in the ApoE gene regulation, as regards the ERα implication in the upregulation of ApoE gene expression and the opposite role played by ERβ. These mechanisms have been observed in studies focusing on specific hormonal therapy in vitro and in preclinical studies [35]. In conclusion, acknowledgment of the exact dimorphic mechanisms of this neurodegenerative disease may lead to more promising methods for treatment, for instance, with a more thorough testing of the biased corticotropin-releasing factor (CRF) receptor coupling [36].

Parkinson’s disease. Parkinson’s disease (PD) is another neurodegenerative disorder mainly characterized by Lewy bodies accumulation (due to α-synuclein aggregation) and depleted dopaminergic (DAergic) neuronal circuits in the nigrostriatal system. Sex differences have been noticed in the incidence and prevalence of the disease as well as in its clinical manifestation. Epidemiologic studies have revealed that men are more vulnerable to developing PD with the male-to-female ratio being 1.6:1. Men also develop a quicker onset of symptoms that include hypersalivation, sexual dysfunction, and excessive daytime sleepiness, but also worse neuropsychiatric and motor symptoms, such as rigidity, rapid eye movement (REM) sleep behavior disorder, and dementia in comparison with women. Women, on the other hand, although reporting more difficulty in daily activities and more frequently symptoms such as fatigue, depression, and tremor, seem to have a relatively better clinical phenotype [37].

The neuroprotective role of female sex hormones has been called into question in the pathogenesis, progression, and treatment response in PD. Studies have confirmed that women of young age or women treated with hormone replacement therapy show high levels of the DAergic system in the nigrostriatal pathway and in monoamine oxidase (MAO). In addition, tomographic imaging in the female brain—even of older women—points out a higher activity in dopamine transporters (DATs) in the striatum compared with the male brain [38]. Recent observations in *Drosophila*, rodents, and humans have demonstrated a regulatory interconnection between the vesicular glutamate transporter (VGLUT) involved in the pathogenesis of the PD and brain vulnerability due to age-related neuronal loss. Results suggest that a greater expression of VGLUT, which increases with age, is present in women, feasibly explicating their resilience to early DAergic neurodegeneration and their lower locomotory symptomatology in comparison with men [39]. Postmortem studies have revealed disparities in the gene expression of PD biomarkers, such as α-synuclein and PINK1, which seem to present a higher mutation rate in men, thus making them more susceptible to neuronal degeneration and oxidative damage.

ER mutations have also been studied, seeking a correlation between ERβ gene mutations and PD onset, but results are still equivocal [40]. A series of mutations in the gene encoding leucine-rich repeat kinase 2 (LRRK2) has revealed a dimorphic pattern, especially in association with urate levels that offer protection. Women and healthy individuals have higher urate levels and unveil fewer LRRK2 mutation patterns compared with men with PD [41].

To sum up, symptomatic treatment of PD for both motor and nonmotor manifestations shall be designed with particular attention to sex-specific aspects. Motor PD symptoms are counteracted by treatment with levodopa. Long-term use of levodopa, however, is associated with the development of on-and-off fluctuation patterns and dyskinesia when the drug reaches high plasma concentration levels. Levodopa’s highest bioavailability has been assumed to be reached more rapidly in women, thus reducing the time span between the initiation of treatment and the onset of levodopa-induced dyskinesia: the mean time interval has been reported to be 4 years versus 6 years in men [41,42].

Epilepsy. Sex differences are eminent in various neurologic disorders, including different types of epileptic syndromes. Sociocultural sex disparities (e.g., due to cultural habits often underestimating mental diseases) along with biological factors tend to influence the incidence and the prevalence of the epileptic spectrum but also interfere with treatment approaches as they might attenuate drugs’ adequacy and safety [43]. Studies from developing countries suggest that epileptic seizures, both provoked and unprovoked, report a higher prevalence in men (50.7 cases per 100,000) than in women (46.2 per 100,000) [44]. Symptomatic partial seizures are also prevailing in men, while idiopathic generalized epilepsy (IGE), juvenile myoclonic epilepsy (JME), temporal lobe epilepsy (TLE), and idiopathic generalized tonic-clonic seizures are found to be more frequent in women [45,46]. In particular, with respect to IGE, the prevalence has been found to be higher in younger women and to decrease with age, assuming a role of sex hormones in the onset. In this regard, numerous studies have agreed that circulating sex hormones and neurosteroids influence the inhibitory GABAergic synapses and the excitatory glutamatergic transmission that are greatly involved in seizures. Analytically, the female sex hormone progesterone is considered to have anticonvulsant properties by negatively modulating the glutamatergic signaling pathway and assisting in the enzymatic conversion of the neurosteroid allopregnanolone, which then can positively regulate GABA transmission by allosterically activating GABA*_A_* receptors. Progesterone’s metabolic products also seem to have an antiepileptic action. Estrogens, on the contrary, especially estradiol, tend to promote seizure incidence, but studies on their proconvulsant performance are controversial: while some studies in rats note an epileptic chart in correspondence to the exchanging sex hormone levels during the menstrual cycle, others suggest the protective role of estradiol. Lastly, androgens also exert a dual action on seizure vulnerability. The metabolism of testosterone in estrogens has proconvulsant effects, but testosterone itself, when metabolized to androstenediol, has similar properties to allopregnanolone [47,48].

Concerning the use of antiepileptic drugs, most studies estimate no significant adjustments of sex hormones upon initiation of treatment; however, lower testosterone, LH, and FSH levels have been observed in patients treated with valproate, which seems to also have a link with endocrine disorders, including amenorrhea, polycystic ovaries, and decreased libido. Consequently, caution in selecting anticonvulsant drugs in case of conception planning must be taken [49,50].

Depression. Most psychiatric disorders, albeit still not entirely decoded, appear to have a link with sex hormones. Depressive disorders, including major depressive disorder and other depressive-like conditions, have different incidence, clinical manifestation, and treatment choices between men and women, evincing the strong pathophysiologic influence of sex hormones [51]. Studies support that women are more likely to develop depressive symptoms upon adolescence and that the risk for depression also increases in case of prior abusive experiences during childhood [52].

The fluctuations of female sex hormones during the menstrual cycle seem to provoke changes in the severity of depressive symptoms. Clinical studies’ results reveal that women with lower progesterone levels during the luteal phase of their cycle experience worse mood symptoms compared with those during the follicular phase. The same results came from a study of women under treatment with contraceptive pills who experienced sleep disturbances and depressive symptoms [53].

Sex hormones also seem to interfere with the hypothalamic–pituitary–adrenal axis (HPA), which is overactive in depressive patients with corticotropin-releasing factor (CRF) being hyperexcreted from the hypothalamus, inducing the release of adrenocorticotropic hormone (ACTH), which results in elevated cortisol level secretion. Studies analyzing the cortisol levels in both depressive and healthy individuals have reported a critically increased concentration of cortisol in depressed women compared with depressed men in comparison with a healthy population [54]. Premenstrual syndrome, pregnancy, postpartum, and menopause are all events of women’s live where sex hormones act immensely [55]. Estrogens have been investigated for their role in the serotonergic, dopaminergic, glutamatergic, and GABAergic system via ERα, ERβ, and GPERs, and their action is deemed to be comparable to that of antidepressants and atypical antipsychotic drugs to a great extent. They have been shown to be positive regulators of tryptophan hydroxylase (TPH) (a 5-HT precursor enzyme), modulators of specific 5-HT receptors, and inhibitors of metabolic catalyzers such as MAO. They specifically block the 5-HT1A autoreceptor and boost 5-HT concentration on the synaptic cleft, also blocking the 5-HT reuptake presynaptically [56]. Ostensibly, treatment of depressive symptoms with antidepressants might benefit by a combination therapy with hormones, as observed in studies carried out on women during postmenopause, which showed improved results when treated with estradiol [57].

Comparative data between AD and PD and other diseases described above are briefly reported in Table 1. Note that the incidence, some clinical aspects, and the outcome show sex-related differences. Treatments also display some sex disparity.

### 3.2. Sex Hormones and the Immune System

Large differences are asserted when comparing the immune system between men and women. Immunological responses vary between the sexes in response to both endogenous and exogenous antigens, creating variations in the incidence and severity of both infectious and autoimmune diseases. Apart from environmental factors that may enlarge this disparity, distinct biological features seem to worsen or protect the health status of individuals. These traits concur with pathogenetic mechanisms as well as with the different response mechanisms for treatment with implications in the clinical practice (e.g., in the use of antimicrobial agents and vaccines. External factors, including choice of food intake, everyday habits, and seeking treatment, are personal decisions of the genders, which expectedly influence immunological responses.

Overall data from available literature indicate a relatively stronger immune system activation and performance in women. This explains their greater capability of developing a better response to infectious agents and the more frequent onset of autoimmune disorders compared with men. The understanding of the molecular mechanisms involved supports a personalized approach in the prevention of the spread of contagious diseases and in facing noncommunicable diseases with a strong immunological component [58,59].

Hormone modulation of immunity. Sex hormones, including estrogen, testosterone, and their receptors, or other genetic factors are biological variables also competent in modifying the signaling pathways of the immune system [58]. These variations have been observed in both the innate and the adaptive immune mechanisms. More accurately, in the first line of the defense system (the innate response), male sex hormones—most importantly testosterone—have been reported to have suppressive effects in the production and activity of numerous immune-associated cells. Results from the majority of in vivo and in vitro research prove the exacerbating effects of testosterone in the immune system’s sensitiveness. Recent experimental evidence, however, suggests that testosterone’s actual role is immunomodulatory rather than exclusively suppressive [60]. For instance, monocytes, which are cytokine-producing cells that can differentiate in macrophages, are incoherently regulated by testosterone, which enhances the release of IL-12 and IL-1 and thus mediates CD4+ helper T-cell differentiation and types of adaptive responses [61]. Preclinical studies in gonadectomized mice treated with testosterone have shown a decreased expression of a distinct type of a pattern recognition molecule (PRM), the toll-like receptor 4 (TLR4) on macrophages’ cell surface, in comparison with castrated mice, reflecting the suppressive abilities of circulating testosterone [62]. Furthermore, testosterone affects TNF-a, IL-1β, and IL-6 expression, as seen in in vivo studies on men under testosterone replacement treatment who develop limited proinflammatory biomarkers [63]. The broad effects of testosterone on the immune response have also been detected by a clinical study on Tsimane men (indigenous people of lowland Bolivia) treated with selective T-cell mitogen phytohemagglutinin (PHA) and B-cell and monocyte mitogen lipopolysaccharides (LPSs). Results show a greater immunosuppressive action of testosterone on T-cell cytokine production upon the use of PHA, while its action after LPS stimulation is not significant [64].

On the other hand, female sex hormones, including estrogen and progesterone and their receptors, are hormonal mediators that contribute to a reinforcement of immune responses, affecting diverse immune cells. Different preclinical experiments and clinical data analysis suggest the association of female sex hormones with the concentration and activity of neutrophils, white blood cells that produce chemotactic messengers and recognize foreign bodies by the expression of humoral pattern recognition molecules (PRMs). More precisely, neutrophil concentration has been proved to be elevated in samples of pregnant women and in women during the luteal phase of their menstrual cycle, underlining the amplifying consequences of estrogen and progesterone in the immune response [65]. Monocytes, per contra, seem to be subdued by high levels of estrogens, which stimulate other proinflammatory responses of macrophages, including the differentiation of CD4+ helper T-cells, which regulate the different types of responses in adaptive immunity. IL-1 and IL-6 cytokines deriving from Th17 CD4+ helper T cells are reported to be elevated in response to estrogens, while IL-17, IL-22, and IL-23 have the opposite modulation. Dendritic cells have also been associated with estrogen concentrations as T-helper 2 type cytokines IL-4, IL-10, and IL-13, which also mediate MHC II expression and trigger B-cell antibody production, are positively affected. Females’ estrogen-induced downregulating actions are evident especially in natural killer cells responsible for producing Th1 cytokines, IL-2, and IFN-*γ* involved in the mobilization of macrophages and cytotoxic T-lymphocytes [61,66]. Consequently, cells and mechanisms involved in the adaptive immune response also seem to be markedly modulated by the circulating female sex hormones. The increased humoral and type 2 responses in the presence of estrogen denote the enhanced B-cell differentiation and antibody production in women. This pattern provides a more secure environment in case of infection but increases the susceptibility to autoimmune diseases [67,68].

Autoimmune diseases. The hyperactive immune system of women is charged to drive the higher incidence of autoimmune disorders in comparison with males: approximately 85% of patients with autoimmune disorders are, in fact, women [69]. In autoimmune disorders, abnormalities in the functioning of the immune responses cause damage to healthy cells. This malfunctioning is influenced by the endocrinological changes that women experience during lifetime (i.e., in puberty, pregnancy, and menopause, when female sex hormones drastically fluctuate) [70]. In particular, autoimmune disorders, including systemic lupus erythematosus (SLE), multiple sclerosis (MS), and rheumatoid arthritis (RA), are known for their elliptical course of manifestation during the different stages of the female reproductive path. For instance, SLE may be heightened during pregnancy, when the previously prevailing Th1 immune response is surrogated by Th2, as imparted by observations in the TH1/TH2 ratio in pregnant women [71]. Conversely, MS and RA present a diminutive manifestation of symptoms during pregnancy since they depend on a Th1-type immunity [69]. SLS is experimentally proved to be associated with estradiol acting through ERα, while ERβ is considered to have minor immunoprotective properties in autoantibody generation by B-cell dysfunction. Further clinical implications between estradiol levels and SLS are spotted in women using contraceptive pills and under hormonal therapy. In the former case, SLS risk has been reported to be determinedly elevated, while, in the latter, opinions are still controversial [72]. In MS, patients present autoantigens, such as myelin’s structural components, which evoke autoinflammatory responses, leading to neurodegeneration. Estrogens, progesterone, and prolactin, with their defending mechanisms on the central nervous system, seem capable of influencing the progression of the disease [73]. Focusing on RA, the association between the concentrations of female hormones and RA risk has also been validated by recent data following women after the first trimester after birth. The disease has shown an aggravation correlated to lower levels of estrogens, progesterone and humoral immune responses, and higher levels of TNF-α and IFN-*γ* [74].

Infectious diseases and inflammatory responses. The different patterns observed in the specific mechanisms of immunity correspond to the dimorphic phenotype of prevalence, progression, and treatment of infectious diseases between the sexes. The course of a pathogen, for example, the replication of a virus, is dependent on the inflammatory mechanisms of the individual that, as mentioned above, show large sex-specific differences and is strongly influenced by hormonal patterns. The hormonal factors addressing these differences have often been investigated in experimental studies on infectious diseases. However, reviews on the mortality rate of various infectious agents underscore the need to better explore the sex-related trajectories of a pathogen’s fate inside the hosts’ cells [75]. Urinary tract infections (UTIs) for instance, are more frequent in women but have greater severity in men. Estrogens, through their impact on the inflammatory responses, have been reported to have an advantageous impact in facing UTI infections, as shown in preclinical studies involving ovariectomized mice that show decreased host resistance after *E. coli* infection and clinical models on postmenopausal women treated with hormones that demonstrated a decreased intermittent pattern of UTIs [76]. Studies analyzing influenza, human immunodeficiency virus (HIV), and hepatitis C virus (HCV), for instance, have shown a higher vulnerability of pregnant women in developing severe influenza infection but also a better management of HCV and HIV-1 progression in women in comparison with men [77]. According to an evaluation of a statistical analysis on the Australian population, it was also stated that a greater number of confirmed influenza A cases were observed in adult women in comparison with men [78]. This was also true for HIV cases, with women having 1.62 times higher prevalence of infection [79]. In contrast, HCV infection is significantly greater in men in comparison with women, according to studies conducted in European and US populations [80].

Hormonal implications in COVID-19. Considerable differences between men and women have also been reported in the coronavirus disease 2019 (COVID-19) epidemiology caused by severe acute respiratory syndrome coronavirus 2 (SARS-CoV-2). A meta-analysis performed on 57 distinct epidemiologic reports stated a 55.00 pooled prevalence of risk of contracting COVID-19 in men in respect to 45.00 in women, with varying susceptibility according to age [81]. However, COVID-19 cases all over the world appear not significantly different between the sexes, whereas mortality rates due to the disease are significantly higher in males. For example, according to data extracted from the Italian National Institute of Health, men accounted for 60% of the total SARS-CoV-2-induced deaths registered in Italy until 20 May 2021, regardless of the age range [82]. In the same vein, data from various geographic regions have reported similar differences. This disparity appears apparently due to both sex (biological) and gender (sociocultural) reasons. For instance, smoking, which enhances the expression of angiotensin-converting enzyme 2 (ACE-2), the binding site of SARS-CoV-2, was reported to increase the risk of severe symptoms of COVID-19 by 1.4 in comparison with nonsmokers [81]. Since smoking habits are more frequent in men, this could represent a relevant gender bias. As concerns biological factors, several studies are underway. Additionally, some points seem to emerge. For instance, sex hormones and sex hormone receptors should be evaluated in the considerations about viral replication and inflammatory response in COVID-19. The above-discussed anti-inflammatory effects of estrogens may play a protective effect in COVID-19 progression. By upregulating TLR signaling and cytokine production, including IFN-α, estrogens were demonstrated to be correlated with less severe cases, as seen in studies in premenopausal women and men of the same age. Contradicting reports also exist about estrogens’ action upon ACE-2 expression and its possible activity in lowering the available binding sites for SARS-CoV-2 [83]. The mechanisms of regulation of ACE-2 receptors by estrogens are in fact under investigation also as adjuvant therapy for the disease [84]. Many studies on animals support the protecting role of estrogens by observing worse COVID-19 outcomes in infected female mice undergoing ovariectomy or treated with estrogen receptor inhibitors [85]. The improving or deteriorating progression of the disease but also the susceptibility to infection have been markedly ascribed to the levels of female hormones present during the lifespan even if the process is still not fully clarified. For instance, pregnant women demonstrate a decline in CD4+ helper T cells and CD8+ T cells but also an upregulation in the expression of ACE-2, which may designate them as more vulnerable to infection and further complications. On the contrary, the reduced circulating and locally produced estrogen levels in menopausal women along with the production of high levels of IgG antibodies early during infection seem to conserve a substantial defense response in comparison with men of the same age [86]. Androgens’ modulation in the immunological response to COVID-19 seems to be likewise in conflict. Studies on infected male mice have claimed a higher viral load and more inflammatory monocytes and macrophages [85]. These results could potentially correspond to the so-called ‘cytokine storm’ image present in most severe COVID-19 cases, where the hyperactivity of the immune system results in elevated concentrations of IL-6, IL-1, TNF-α, IFN-*γ*, and other proinflammatory cytokines [87]. Further scientific research on male animal and human candidates have led to the assumption that testosterone is associated with COVID-19 mortality, but its role is still debated. Scientific assumptions divulge the positive modulatory effects of testosterone on ACE-2 receptors and on transmembrane protease serine 2 (TMPRSS2), whose action is to prime spike proteins of SARS-CoV-2 [88]. How the elevated testosterone levels exacerbate the natural course of the disease has also been established by control studies on prostate cancer patients. Although immunocompromised patients endure COVID-19 more severely, prostate cancer patients treated with hormonal modulators used to abate androgens and hence TMPRSS2 are less imperiled by the disease [89]. Recent studies conjecturing the link between the increased risk of severe COVID-19 and androgen concentration have found a higher risk of hospitalization in men with androgenetic alopecia, a common dermatological condition that is characterized by hyperandrogenicity and hyperactivity of androgen receptor genes [90]. A shorter length of the CAG trinucleotide repeat in the first exon of AR genes, an alteration that is common in males with this type of disorder, has been linked to more severe and fatal COVID-19 cases [90]. Another noticeable clue of the effects of testosterone was found in a study on hospitalized patients with mild and severe COVID-19, where severely ill male patients reported lower concentrations of testosterone [91]. Decreased testosterone levels were also associated with endothelial cell dysregulation and production of reactive oxygen species as SARS-CoV-2 competes with angiotensin II for binding to ACE-2 receptors, causing an increase in the angiotensin II levels and a decrease in angiotensin-1–7, an important vasoprotector of the renin–angiotensin system [92]. The data provided thus fur may encourage the use of testosterone in male patients by attenuating the proinflammatory state present in testosterone-deprived men without impeding the capital immune responses while also reinforcing the respiratory system [93]. In conclusion, sex hormones’ influence on COVID-19 is indisputable but not yet fully understood. Awareness of how they impact the course of the disease may implement novel treatment strategies as underlined by the increased interest in sex differences in possible target treatment recently shown by researchers involved in drug development in COVID-19 [94].

The scenario described above is briefly summarized in Table 2. It is well known that the occurrence of autoimmune diseases is higher in women (up to 9:1). Some clinical aspects, mainly associated with pregnancy, and some biological features are also indicated. As concerns infectious diseases, only some examples have been provided. Of note, COVID-19 severity and lethality are significantly higher in men.

### 3.3. Sex and Gender Implications in Cardiovascular Diseases

Cardiovascular diseases (CVD) belong to those medical conditions that show major sex-related differences. Sex dimorphism has been recognized in the risk, progression, outcome, and management of different heart and blood vessel pathologies, including heart failure, atherosclerosis, hypertension, and stroke. Once again, genetic factors (e.g., the partial inactivation of the X female chromosome, sex hormone concentration, and their receptors’ location) seem to play a role in the disparity depicted below. 

Stroke. Stroke is an emergency condition that requires immediate medical treatment to avoid the worst outcomes. It consists of two possibilities of clinical manifestation: an ischemic stroke by blood clot formation (more frequent in women) or a hemorrhagic stroke. Stroke may often lead to a progressive neurodegeneration correlated to the extent of damaged tissue [95]. Stroke prevalence and mortality are both higher in women (1.5% more deaths in women of all ages and 3.3% prevalence, while 2.7% in men). These numbers, however, are greatly dependent on the exceeding incidence of stroke in elderly women, which is lower in younger ages [96]. Prognosis after stroke’s incidence in women has also been found to be rather poor mainly due to defects in the carotid repair mechanisms—which may be caused by hormonal regulatory effects—including that on atherosclerotic plaque formation, but also anatomical differences, such as a smaller diameter of blood vessels in women [97]. Experimental studies have been conducted in order to enlighten the biological risk factors involved in the main different types of strokes (ischemic and hemorrhagic). Studies on mice have suggested that brain injury after ischemia is prevailing in young males rather than females and that this trend is inverted after the age of 15 months. This finding insinuates that the age-progressive loss of estradiol would be a possible explanation, but still, the similarly low levels observed before puberty hamper a clear view of the exact hormonal regulation. One putative explanation is that hormones are responsible for permanent organizational effects, rather than “activational” effects, having regulatory actions on sustaining stroke-induced brain damage. It is experimentally proved on rodent models that estradiol may decrease brain damage from a stroke in both sexes, as seen in OVX females that have lost this neuroprotection and in males treated with estradiol after a stroke incidence that, on the contrary, have gained it [98]. Animal studies’ results have also revealed that sex-specific ischemic sensitivity may not be consistently related to genetic alterations, from neither the extra female X chromosome nor the specific genes on the Y chromosome. Regarding testosterone in younger males, studies enunciate an elevated risk of developing CVDs. Orchiectomized male rats treated with testosterone have confirmed testosterone’s predisposing effects. Ultimately, postischemic stroke patients have also been found with low testosterone levels. This could be due to the massive conversion of testosterone to estrogen (i.e., to the intrinsic protective mechanisms of aromatization of testosterone) [99].

Obesity. Obesity is one of the main risk factors for developing CVDs in both men and women. The abnormal body composition in obesity, reflecting in excess fat mass but also lean weight, may cause heart failure and coronary artery disease, but may also alter the normal anatomy of the myocardium, leading to cardiomyopathy. The main mechanisms leading to these comorbidities rely on the adipose tissue production of proinflammatory cytokines and favoring atherosclerotic plaque formation [100,101,102]. Hormone levels may influence the pathophysiology of the obese phenotype and thus the regulatory pathways, leading to CVDs. Worth noting, the so-called ‘obesity paradox’ states that the obese status, especially in the elderly, mitigates the outcome of an already-existing advanced form of heart failure, meaning that an increased body mass index may minimize mortality rates. Although a minimum number of studies have been conducted investigating this paradox in both sexes, the same trend has been reported in both women and men [103]. In general, obesity is an ongoing inflammatory state with metabolically active processes, including an excess adiposity and secretion of adipokines, but also a higher insulin secretion in comparison with lean subjects, an increased insulin resistance, and elevated production of very low-density lipoproteins (VDLs). Several studies have underlined the association of estrogens to these processes. For instance, postmenopausal women show an excess in adipokines and an amplified generation of immune mediators, leading to a higher risk of cardiometabolic disease. This syndrome itself involves insulin resistance, dyslipidemia, and hypertension as its clinical symptoms, augmenting the risk for coronary artery diseases, heart attack, and stroke [104]. Nonetheless, statistical data from different European countries declare a higher prevalence of overweight and obesity in adult men in comparison with women of the same age groups, although data seem to differ according to the geographical position and social interactions of the individuals [105]. On a biological point of view, due to the ability of the adipose tissue to secrete adipokines, such as the proinflammatory leptin and the anti-inflammatory adiponectin and resistin, it may be considered an organ with endocrine functions, which may contribute to CVDs, cancer development, and prognosis (e.g., in human epidermal growth receptor 2 (HER2)-positive breast cancer) [106,107]. Sex differences have also been reported in the adhesion of patients to attainable weight loss programs; hence a better comprehension of this interrelation would promote more proper diet method selections [108].

Atherosclerosis. It is a chronic inflammatory state characterized by atherosclerotic plaque formation in different arterial vessels, obstructing blood flow towards the tissues and resulting in ischemic conditions. Over time, the plaques may harden, narrowing the arteries’ diameter, or even burst, as a result of a blood clot formation. Atherosclerosis is the main cause of morbidities and mortality among both sexes, mainly due to cardiovascular events, such as myocardial infraction (MI), heart failure, and stroke. The literature suggests a more vigorous manifestation and poor outcome of atherosclerosis in men. Limited animal angiographic analyses support this propensity, while also bringing to the fore the finding that atherosclerotic inflammation is the one related to worse outcomes and prognosis, rather than the plaques’ volume. Further studies will be decisive in revealing the precise mechanisms of these events, but also in exploring the sex variable in CVDs [109]. Epidemiologic sources, as already mentioned, have demonstrated a higher risk of prevalence of CVDs in young men rather than women, which also applies in atherosclerosis evolution. This may be partly explained by risky behavioral habits, including smoking and drinking alcohol, which are more common among men, but also by the appraised protective mechanisms of estrogens among younger women as documented also in animal studies. What is quite arguable, though, is the safety of hormonal treatment in men and women in older ages.

Although estrogen and testosterone in high quantities lessen the atherosclerotic risk, a potential treatment for deprived older individuals may have deleterious consequences [110]. Atherosclerotic plaque formation has been vastly scrutinized for its composition and inflammatory state, without, however, inspecting the issue of sex-related differences. A few notions have been affirmed regarding plaque’s morphology, with a thicker fibrous cap in stabilized plaques in younger women and a greater necrotic core in older ones. A greater risk of plaque rupture due to high total cholesterol levels has been recorded, compared with plaque erosion, which is present in broader, less calcified arteries of young women [111]. Sex differences have been noted not only in the compositional elements of atherosclerotic plaques but also in the active inflammatory functions. More precisely, according to a source examining the inflammatory steps in myocarditis, testosterone could be responsible for triggering mast cells and macrophage activation, enhancing TLR-type macrophages and foam cell aggregation within the plaques, inducing structural rearrangements and causing a subsequent thrombotic MI. Conversely, women’s estrogens have been associated with an elevated quantity in antibodies and autoantibodies, along with the ones against oxidized LDL, which settle on the walls of the narrow blood vessels, producing thrombosis and MI [112]. In summary, sex-specific differences have been found in the risk for developing atherosclerosis and in the morphophysiological elements of the plaques.

Hypertension. Hypertension is a major risk factor for CVDs that shows gender and sex-specific patterns. Studies report a higher prevalence of hypertension in men until late adulthood, when the proportions reverse. Again, menopausal hormonal alterations seem to be involved in the exceeding hypertension rates in elder women [113]. Studies on the monitoring of arterial blood pressure (BP) adjustments during the menstrual cycle and on a 24 h scale (pressure Holter) of both hypertensive and healthy subjects have demonstrated notable variations between the sexes. Women appeared to have higher BP during menstruation and on the follicular phase rather than the luteal phase of their menstrual cycle [114]. Although distinct clinical symptoms between men and women have been conceded, adaptations on guidelines for keeping BP under control are not sex specified. Optimal BP management would be achieved by integrating more women in clinical scientific measurements and, more importantly, in clinical trials since response to treatment may vary between the sexes [115].

Heart failure. Heart failure (HF) represents one of the most common CVDs affecting both men and women globally. Sex diversity applies to risk factors contributing to its incidence and its phenotypic spectrum, which differs markedly, among others. Hyperinflammation, arterial vessel rigidity leading to vascular stiffness, hypertensive disorders in pregnancy, emotional stress, and breast cancer treatment are some of the risk factors implicated in HF in women that differ from those suspected in men [116]. The hallmarks of HF’s sex-related picture include an overall greater risk of men suffering from reduced ejection fraction (HFrEF) or midrange ejection fraction (HFmrEF), while women have more preserved ejection fraction (HFpEF). Women seem to be afflicted by more coexisting comorbidities but have a more favorable prognostic and mortality rate than men [117]. The different combinations of comorbidities are related to separate HFpEF phenotypes; one characterized by a decreased cardiac remodeling and reduced natriuretic peptide hormone release; one with obesity-related clinical indexes; which are more frequent in women; and a more severe one associated with chronic kidney failure, left ventricular remodeling, and a less auspicious prognosis, more observed in men [116]. The aftermath of a HF diagnosis results, altogether, in a lower quality of life in women despite having a more optimistic attitude, whereas men are more dramatically impacted by the psychophysical boundaries [118]. In summary, behavioral characteristics along with other socioeconomic factors (e.g., higher attention and reporting of women to physicians) concluded in a similar hospitalization rate between men and women, despite the most severe HFrEF phenotype affecting men [119]. Treatment of HF also revealed altered efficacy and a high adverse reaction rate, particularly in women. Management of HF and other CVDs may be improved by orienting applicable studies towards a more gender-analogous perspective [120].

The scenario described above is briefly summarized in Table 3. Some risk factors for CVD are described here together with some features of myocardial infarction. In particular, occurrence, clinical aspects, and some notes on sex differences as concerns therapy and biological features are reported.

### 3.4. Sex-Specific Exogenous and Endogenous Factors in Oncology

In the oncologic field, sex disparity has been investigated in either epidemiological or mechanistic studies. Several sex/gender differences have been discovered, highlighting that sex and gender could be critical factors able to influence the predisposition, the response to treatment, and the mortality of multiple types of cancer [121]. The mechanisms underlying this disparity involve genetic and epigenetic mechanisms (e.g., acetylation, myelination, and circulating sex hormones themselves seem accountable for “acute” or “chronic” hormone-dependent cancer types) [122].

Sex-biased cancer epidemiology. Bladder, kidney, colorectal, liver, head, neck, brain, skin, and hematologic are types of cancer more prevalent in men, while breast, thyroid, cranial nerve, and a few types of digestive system cancers are noticed with a higher incidence in women [123]. Apart from impacting the occurrence and development of different cancer types, sex disparity may also influence mortality rates. According to clinical data, the mortality rate in men is 1.43 times higher than that in women for a great number of cancer sites. The male-to-female (M/F) ratio’s range is determined also considering the age factor, including menopause. In particular, worldwide reports reveal a higher probability of mortality in men for the following cancers: bladder (M/F ratio): 4.12; colorectal: 1.5; larynx: 5.17; and hypopharynx: 5.75, whereas mortality is higher in women for the following cancers: thyroid: 0.33; anus: 0.85; and gallbladder and biliary tract: 0.94 [124,125]. Furthermore, what is noticeable, is the rate of survival in lung and bronchus carcinomas in women, which significantly exceeds that reported in men. A great number of trials have been carried out over the years, analyzing the correlation of sex disparity and carcinogenesis, but frequently, the etiology has been poorly investigated. Sex differences in cancer incidence and response to treatment need specific analyses. Clinical trials evaluating sex-specific pharmacokinetic differences appear as mandatory. Non-sex-related cancers, e.g., prostate or cervix, must obviously be considered as biologically distinct [126]. In addition, the sex-related interplay between a tumor microenvironment, including estrogen and MicroRNAs, and gender disparity in cancer has more recently been underscored [127,128].

On the external stimuli. Beyond biological differences, different patterns in environmental risk factors can help to explain the different epidemiological scenarios between the two sexes. A general conception that might partially explain the differences in cancer incidence and mortality rates between men and women is the fact that women tend to seek medical assistance and to perform routine controls more often in comparison with men. This means that they might get an earlier diagnosis and treatment, while men might get diagnosed in later stages of the disease [125]. Another indicative paradigm of sex influence in oncology is represented by numerous studies analyzing the effects of tobacco smoking on lung cancer in men and women. Up-to-date data suggest that the incidence of lung cancer in men, which was formerly higher, is now almost changeless, whereas it has been found to be increased in women who are increasingly developing smoking habits [129]. Women are presumed to be more susceptible to developing lung cancer induced by smoking, as elaborated by preclinical and clinical studies that explore the influential role of sex hormones. Lab female mice that underwent ovariectomy show less lung cancerous conversions, while castrated males report an elevated risk. In human populations, women smokers present a higher number of oxidative damage-causing DNA adducts correlated to tobacco carcinogens. Moreover, female smokers treated with a combination therapy of estrogen and progestin, as well as male-to-female transsexuals treated with hormonal substitution therapy, manifest a greater risk of developing lung cancer. An attainable justification may be given surveying the suppressive effects of female sex hormones on the metabolic pathway of the main carcinogenic compounds found in cigarettes, which leads to a declined clearance [130,131,132].

Furthermore, a noticeable difference between the sexes occurs when taking into consideration ultraviolet (UV) light exposure. UV is another worth-mentioning risk factor that accounts for specific types of skin cancer and exhibits vast variations between men and women. Clinical studies on Caucasian populations have demonstrated an association between cutaneous melanoma and basal cell melanoma and UV index in young women but not in menopausal women; thus, female hormones’ regulatory effects on mutagenesis may be a valid hypothesis. On the contrary, the risk for this event in men gradually increases with age [133]. A potential explanation may be provided by further studies examining sex hormones’ role in melanoma progression. Animal models in mice revealed that estradiol has stalling effects on sunlight-induced immune suppression. The role of the specific ER subtypes also seems to attain tumorigenic and metastatic regulatory functions. More specifically, the protumorigenic and antitumorigenic attributes of ER and ER, respectively, have been acknowledged [134]. In summary, the fact that women tend to use more photoprotective cosmetic products on their skin and the biological structural differences between male and female skin layers should also be considered, but further studies are imperative to clarify the enigmatic involvement of the age axis [135,136].

Genetics and sex hormones in specific types of cancer. Inheritance of specific genes deriving from X and Y chromosomes is an evident factor, suggesting that genetic sex disparities engage in the onset of multiple types of tumors [123]. Scientific literature so far has shown the protective mechanisms of the random inactivation of one of the X chromosomes, normally occurring in women, against multiple cancer types. In this way, mutations in oncogenes or tumor suppressor genes might be avoided, while in men, they might be ex- pressed. Additionally, as this inactivation is incomplete, women have a higher probability to express “escape from X-inactivation tumor suppressor” genes from both alleles, which results in a diminished carcinogenic occurrence [121], while in men, mutations in many of these tumor suppressor-specific genes manifest at a higher probability [123,137]. The presence of another perplexing RNA gene, the X-inactive-specific transcript gene (XIST), which is predominately expressed in women and is essential for X inactivation, has been shown to have a modulatory action on tumor suppressor breast cancer-associated gene 1 (BRCA1). It is also suggested that it augments the susceptibility of lung cancer, non-Hodgkin’s lymphoma, and testicular cancer in men [138]. Valuation of another gene, BRCA2, is also crucial as mutations of this gene have been correlated to male breast cancer to a high extent [139]. Moreover, recent studies have shown a relation between a newly discovered prolactin receptor type, the human prolactin receptor intermediate isoform (hPRLrI), and breast cancer development. Results have demonstrated the interactive pathways of the transcription factor STAT5, which is responsible for the production of prolactin, and ERα, able to influence breast cancer progression. Such studies launch potential research on novel therapeutic biomarkers [140]. Circulating sex hormones themselves are implicated in the carcinogenesis process. More specifically, prevailing in women, breast and ovarian malignancies are types of cancer that may be provoked by modifications of ER*α,* ERβ, progesterone receptor (PR), and HER2 [141]. ERα and ERβ as well as AR are involved in the development of tumorigenesis in multiple tissues. For example, ERβ has been found to be responsible for modulating the NOTCH1 gene associated with squamous cell carcinoma [121]. Ultimately, gonadal sex hormones have also been recognized as having a regulatory role in angiogenesis and metastatic states of cancer. Estradiol has been associated with elevated production of endothelial progenitor cells (CD34+, VEGFR2+), endothelial nitric oxide synthase (eNOS), and promotion of endothelial cell propagation, through ERα and ERβ binding. On the contrary, androgens have been recorded to enhance the propagation of endothelial cells by regulating angiogenesis-linked genes, such as HIF-1a [142]. However, this disparity still deserves further studies before reaching the patient’s bed. 

Indirect effects of epigenetic alterations on gene expression. Cancer susceptibility in men and women may also be influenced by the epigenetic changes that modulate the expression of specific genes. DNA methylation, for instance, has been widely studied for its regulatory actions on gene expression [143]. Sex disparity is characterized by a certain difference in the methylation sequence of CpG sites. This variation, when present in crucial regulatory genes, is enough to play a noteworthy influence on cellular response upon tumorigenic hazards [123]. DNA methylation is evidently regulated by sex hormones that may alter the methylation pattern between men and women [144]. A high percentage (85%) of more than 11,000 CpG sites display increased methylation in men than in women according to a study carried out in children at different points of their lives until adolescence [145]. Further studies have substantiated the data above, showing a diversity in methylated genes, as well as in methylated CpG sites, in men and women [146]. This diversity is reported not only in discrete regions of the inactive and active X chromosome, but also in autosomal CpG sites [147]. Studies examining blood long interspersed nuclear element-1 (LINE-1) and Alu have revealed a higher methylation rate in men, while in saliva tissue cells, the rate has been reported to be higher in women [148]. The consequences of this heterogeneity include a different response to external risk factors, as well as a modulation in tumorigenic expression and progression. For instance, hypermethylation in CpG islands in promoter regions of tumor suppressor or DNA repair genes may modify their expression. On the other hand, hypomethylation of CpG is present in tumorigenic tissue [149]. Whole-genome analysis may elucidate more specifically the dimorphic nature of the DNA methylation process.

Sex-specific histone modifications also show a regulatory role in malignant mutations. Genes that encode histone deacetylases (HDACs) and histone acetyl transferases (HATs) tend to show a diversity in their expression patterns. For instance, the KDM6A gene eludes X-inactivation and is expressed by both female chromosomes, allowing a chance of remittance in case of mutation, whereas males carry KDM6C, a homologous gene that encodes for a rather incompetent protein [150]. Moreover, X-linked noncoding microRNAs seem to also exert a modulatory role in the pathogenesis of cancer by interfering with ERα and ERβ expression and autophagic processes among others; more precisely, miRNA-221 and miRNA-222 impede ERα expression, which is also involved in cardiotoxicity disparities in patients undergoing chemotherapy [128]. 

Sex influence in the response to therapy. Sex and gender should be considered determinant parameters in identifying the right type of strategy of therapy for cancer patients. For instance, immunotherapy is putatively influenced to some extent by sex hormones, although data analysis is still limited or vague. The uncertainty of the results might be justified because of the vast number of environmental and molecular factors that need to be taken into consideration when investigating a tumor progression. Meta-analyses investigating sex disparities in the use of immune checkpoint inhibitors (ICIs), including among others anti-programmed cell death receptor 1 (anti-PD1), anti-PD1 ligand (anti-PDL1), and anti-cytotoxic T-lymphocyte-associated protein 4 (anti-CTLA4) antibodies, have resulted in disputable findings regarding treatment and efficacy. It has been stated that ICI treatment alone is more effective in men in comparison with women, plausibly because augmenting the already-elevated immune response present in women does not generate a more drastic therapeutic outcome. However, further studies analyzing the hazard ratio and survival rate in patients treated with ICI and other chemotherapeutic agents have reported a more beneficial response in women than in men. This might be explained by the fact that newly mutated tumor sites induced by the chemotherapeutic agents may increase immunogenicity in women [151]. Additional studies have reported sex-biased patterns of tumor mutation burden and immunogenicity, as well as different immune checkpoints and other biomarkers on several types of cancers [152,153].

Sex disparity is also present in the use of monoclonal antibodies, vaccines, or other tumor destructive agents as sex can modify the pharmacokinetics, pharmacodynamics, and toxicity of several drugs [154]. It has been acknowledged that doses of antineoplastic agents shall be adjusted not only according to the body surface area (BSA), but also accounting for the fat-free mass (FFM), which is considerably higher in women. The assertion of such difference is confirmed as women develop a higher distribution of lipophilic drugs, while men show a higher distribution in water-soluble drugs [121]. Due to the various levels of fat percentage in the two sexes, antineoplastic drugs, such as paclitaxel; fluorouracil; anthracyclines, including doxorubicin; tyrosine kinase inhibitors, including imatinib; and monoclonal antibodies, including rituximab, appear to have a higher elimination rate in men; thus the dose of these drugs shall be modulated accordingly. The different drug concentrations, which may contribute to toxicity, are also generated by differences in the metabolic processes of the organism. Enzymes such as CYP3A, responsible for metabolizing a great number of drugs in the liver, were proved to be more effective in women, whereas drug transporter P-glycoprotein is more active in men [155]. The latter, along with the fact that estrogens are responsible for reducing gastrointestinal motility, may explain the poor oral bioavailability and absorption of drugs in the gut of women [156]. In conclusion, since sex and gender influence the incidence and the natural history of multiple types of cancer, it is paramount to include sex as a variable in the perspective of the development of personalized cancer therapy [157].

The scenario described above is briefly summarized in Table 4. In particular, some sex differences in terms of occurrence, mortality, and response to therapy are reported. The mechanisms involved in this disparity seem to involve environmental risk factors, sex hormones, genetics, and epigenetic alterations.

## 4. Conclusions

Designing medicinal control studies fostering sex- and gender-specific trials appears pivotal in order to develop a more personalized medicine. Sex-biased experimental (i.e., preclinical) and clinical practices should be improved. In fact, females are barely included in preclinical studies on animals (e.g., usually performed on male rodents) and in interventional or observational biomedical studies. This is apparently due to the fact that the female sex clearly shows potential hormonal implications, more challenging to interpret. The current discrepancy exemplifies the need for further, more precise guidelines, which would not only integrate the paradoxical effects of biological features, such as sex hormones, and their eventualities on altering the propensity of a disease or a drug’s pharmaceutical profile, but also acknowledge a nondiscriminative implementation of the whole gender spectrum in medical studies that would permit novel distinctive therapies. Personalized and sex-specific medicine may effectuate revolutionary clinical techniques and therapeutic applications, emblematic of an improved quality of healthcare and medicinal environment. The embodiment of unconventional, nonstereotypical guidelines in biomedical research journals that promote sex and gender equity would be a great starting point in furtherance of attaining the zenith of unequivocal scientific findings without limitations [158]. Finally, to date, policymakers have failed to embrace the gender equality perspective in designing and implementing specific solutions to gender inequalities. To do so, more efforts in accountability and global strategies are needed to acknowledge that gender inequality is not merely confined to health and development, but substantially represents a matter of fairness and social justice.

## Figures and Tables

**Table 1 biomolecules-12-00413-t001:** Sex differences in neurologic conditions, Comparative data between AD and PD and other diseases described in the text are briefly reported. Note that the incidence, some clinical aspects and the outcome show sex-related differences. Treatments also display some sex disparity.

Sex and Neurologic Disease				
	Incidence	Clinical Aspects	Outcome	Therapy
**Alzheimer’s disease**	F > M -Ages 65–69: 0.7% vs. 0.6% [24] -Ages 86–89: 14.2% vs. 8.8% [24]	-Depression and anxiety: F > M [26] -Agitation and hostile behavior: M > F [26]	-Dementia: F > M [25,26] -Cognitive deterioration: F > M [25,26]	Early hormonal therapy: protective for females [34]
**Parkinson’s disease**	M > F -Male-to-female ratio 1.6:1 [37]	-Hypersalivation, sexual dysfunction and excessive daytime sleepiness: quicker onset in M [37] -Neuropsychiatric and motor symptoms: worse in M [37] -Difficulty in daily living activities: F > M [37] -Fatigue, depression and tremor: F > M [37]	-Dementia: M > F [37] -Death: M > F [37]	Mean time interval between initiation of treatment with levodopa and onset of levodopa-induced dyskinesia: F > M (6 y vs. 4 y) [41,42]
**Epilepsy**	M > F: 50.7/100,000 vs. 46.2/100,000 [44] Subtypes [45,46]: -Symptomatic partial type seizures: M > F -IGE: F > M -JME: F > M -TLE: F > M -Idiopathic generalized tonic-clonic seizure: F > M			Affiliation of valproate with endocrine disorders in female (such as amenorrhea, polycystic ovaries and decreased libido) [49,50]
**Depression**	F > M [52]			Women in post menopause might benefit from a combination therapy with hormones [57]

**Table 2 biomolecules-12-00413-t002:** Sex differences in immunological responses. The occurrence of autoimmune diseases is significantly higher in women than in men (up to 9:1). Some clinical aspects, mainly associated with pregnancy, and some biological features are reported. As concerns infectious diseases only some examples have been provided. Of note, COVID-19 severity and lethality are significantly higher in men.

Sex and the Immune System				
		Occurrence	Clinical Aspects	Biological Features
**A** **U** **T** **O** **I** **M** **M** **U** **N** **E** **D** **I** **S** **O** **R** **D** **E** **R** **S**		-F > M -~85% of cases involve women [69]		
	**Systemic Lupus Erythematosus**	-F > M -Higher risk in women using contraceptive pills [72]	Elliptical course of manifestation in females: Heightened during pregnancy [71]	
	**Multiple Sclerosis**		Elliptical course of manifestation in females -Diminutive manifestation symptoms during pregnancy [69]	Protective role of oestrogens, progesterone and prolactin on the central nervous system [73]
	**Rheumatoid Arthritis**		Elliptical course of manifestation in females -Diminutive manifestation symptoms during pregnancy [69] -Aggravation after the first trimester after birth [74]	Lower levels of oestrogens, progesterone and humoral immune responses and higher levels of TNF-α and IFN-γ [74]
**I** **N** **F** **E** **C** **T** **I** **O** **U** **S** **D** **I** **S** **E** **A** **S** **E** **S**	**Urinary tract infections**	F > M [76]	Severity M > F [76]	
	**Influenza A**	F > M [78]	More severe course in pregnant women [77]	
	**HIV**	F > M [79]	Milder progression in women [77]	Severity M > F [77]
	**HCV**	M > F [80]	Milder progression in women [77]	Higher intensity and prevalence in males [77]
	** *COVID-19* **	M > F -Not significant differences (Pooled prevalence 55.00 vs. 45.00) [81]	Severity M > F [82] -Comparing premenopausal women and men of the same age [83] -Mortality associated with testosterone [88] -Higher propensity of hospitalization in men with androgenetic alopecia [90]	-Protective role of oestrogens [83] -Bifold features of testosterone [89,92]

**Table 3 biomolecules-12-00413-t003:** Sex differences in cardiovascular diseases. Some risk factors for CVD are described here together with some features of myocardial infarction. In particular, occurrence, clinical aspects and some notes on sex differences as concerns therapy and biological features are reported.

Sex and Cardiovascular Diseases				
	Occurrence	Clinical Aspects and Outcome	Therapy	Biological Features
**Stroke**	Prevalence: F > M (3.3% vs. 2.7%) [96] Due to the exceeding incidence of stroke in elderly women, which is lower in younger ages [96]	-1.5% more deaths in females of all ages [96] -Weak prognosis: F > M [97]		Defects in the carotid repair mechanisms in females, which may be caused by hormonal regulatory effects and anatomical differences [97]
**Obesity**	Prevalence in young adults in Europe M > F [105]		Different adhesion of patients to attainable weight loss programs [108]	-Excess in adipokines and amplified generation of immune mediators in postmenopausal women [104]
**Atherosclerosis**		Severe manifestation and outcome: M > F [109]		-Protective mechanisms of oestrogens among younger women [110] -Plaque morphology variations with aging in females [111] -Sex differences in the active inflammatory functions of atherosclerotic plaques [112]
**Hypertension**	Prevalence -Until late adulthood: M > F [113] -From late adulthood: F > M [113]	-Higher BP during menstruation & follicular rather than the luteal phase of menstrual cycle [114]	Lack of sex-specified guidelines for keeping BP under control [115]	
**Heart failure**		-Reduced ejection fraction (HFrEF) and mid-range ejection fraction (HFmrEF): M > F [117] -Coexisting pathognomies: F > M [117] -Weak prognosis: M > F -Mortality rate: M > F [117] -Low quality of life after diagnosis: F > M [118]	-Biased efficiency [120] -Adverse reaction rate: F > M [120]	Risk factors in women: -hyperinflammation [116] -arterial vessel rigidity leading to vascular stiffness [116] -hypertensive disorders in pregnancy [116] -emotional stress [116] -breast cancer treatment [116]

**Table 4 biomolecules-12-00413-t004:** Sex differences in oncology. Some sex differences in terms of occurrence, mortality and response to therapy are reported. The mechanisms involved in this disparity seem to involve environmental risk factors, sex hormones, genetics and epigenetic alterations.

Sex and Oncology					
Epidemiology		Susceptibility			Response to Therapy
Occurrence	Mortality	Genetics and Sex Hormones	Epigenetic Alterations	Environmental Risk Factors	
M > F: Bladder, kidney, colorectum, liver, head, neck, brain, skin, hematologic [123]	M > F (1.43 times higher mortality rate in males) [124,125]	-Random inactivation of one of the X chromosomes in females could prevent mutations in oncogenes or tumor suppressor genes [121] -Expression of “escape from X-inactivation tumor suppressor” genes from both alleles in women, results in decreased cancer incidence [121]	Biased DNA methylation pattern (especially in CpG sites), [144] results in different: -response to external risk factors [149] -modulation in tumor expression and progression [149]	Seeking for medical assistance & performing routine controls: F > M [125]	-Efficacy of treatment with ICI alone: M > F [151] -Beneficial response to treatment with ICI and other chemotherapeutic agents: F > M [151]
F > M: Breast, thyroid, cranial nerves & some digestive system cancer types [123]	Probability of mortality M > F in: [124,125] a. Urinary bladder (M:F ratio = 4.12) b. Colorectal (M:F ratio = 1.5) c. Larynx (M:F ratio = 5.17) d. Hypopharynx (M:F ratio = 5.75)	XIST, predominantly expressed in females, determines: a. modulatory action on BRCA1 [138] b. increased susceptibility in males to develop lung cancer, non-Hodgkin’s lymphoma and testicular cancer [138]	Regulatory role of the sex-specific histone modifications in malignant mutations [150]	-Tobacco smoking: Increased risk of lung cancer in women due to the suppressive effects of female sex hormones on the metabolization of carcinogenic compounds of cigarettes [130,131,132]	-Distribution of lipophilic drugs: F > M [121] -Distribution of water-soluble drugs: M > F [121] -Elimination rate of antineoplastic drugs, anthracyclines, tyrosine-kinase inhibitors and monoclonal antibodies: M > F [155] -Oral bioavailability and absorption of drugs in the intestines: M > F (because of reduced gastrointestinal motility caused by estrogens) [156]
	Probability of mortality F > M in the cancer of: [124,125] a. Thyroid (M:F ratio = 0.33) b. Anus (M:F ratio 0.85) c. Gallbladder and biliary tract (M:F ratio 0.94)	-Association of estradiol with promotion of endothelial cell propagation, elevated production of CD34+, VEGFR2+ and eNOS [142] -Role of androgens in enhancing endothelial cell propagation through regulation of angiogenesis-related genes (e.g., HIF-1a) [142]	Modulatory role of X-linked noncoding microRNAs in cancer pathogenesis (e.g., by regulating ERα and ERβ expression and autophagic processes) [128]	-Ultraviolet (UV) light exposure: a. association between UV index and cutaneous melanoma and basal cell carcinoma in young women but not in menopausal women [133] b. stalling effects of oestradiol on sunlight-induced immune suppression [134]	Different drug concentrations also due to differences in the body’s metabolic processes: -greater effectiveness of enzymes such as CYP3A in women [155] -greater activity of the drug transporter P-glycoprotein in men [155]

## Data Availability

The study did not report any data.

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
