# Peer review of "Hormones and Sex-Specific Medicine in Human Physiopathology"

_biomolecules, 2022, doi:10.3390/biom12030413_

Round 1
Reviewer 1 Report
I have no further comments. The authors have revised their review based on the feedback provided.
Reviewer 2 Report
The paper has been improved.
Thank you.
This manuscript is a resubmission of an earlier submission. The following is a list of the peer review reports and author responses from that submission.
Round 1
Reviewer 1 Report
It is an interesting topic for review. However, it is unclear if men and women differ in physiological responses without pathology. This should be stated when known.
Overall, the review is very dense and desultory. It would benefit from reorganization in flow of subject matter.
Title: the title is inappropriate as clinical research does not have sufficient data on “gender” medicine. What the authors probably mean is sex- men versus women.
Line 17: what do authors mean by sex ratio?
Lines 37-39: Race might be skewed in clinical trial recruitments, however, in most mental health-related or neurodegenerative diseases, women are over represented. The authors should make an attempt to highlight whether original trials had equal numbers of men and women. This would be especially pertinent for data for neurological diseases discussed here.
Lines 40-42. Pregnant women and children are a different category and exclusion of pregnant women from clinical trials for drugs is justified. Menstrual cycle is an integral part of women’s physiology and should not be ignored. In that regard, there is diurnal changes in testosterone in men, which are larger than the daily swing in hormones in women, yet that aspect is never taken into account or men are not excluded because of T levels.
Lines 43-47: While the authors make a distinction between sex and gender (lines 43-47, Introduction), they then interchangeably use the terminology. This needs to be corrected throughout the text. If there is information available on gender, then it needs to be specified if that pertains to transmen, transwomen, binary etc, if not, then the information is only about biological sex and should be referred to as men versus women. Male and Female have social implications associated with the terminology but is acceptable.
The authors are directed to the official statement on sex and gender by the Endocrine Society, which might help them in re-organizing this review.
https://pubmed.ncbi.nlm.nih.gov/33704446/
Line 65- when talking about T value, age might be an important factor to state. In that regard, please note that unless mass spectrometry was used, values determined by RIA or ELISA are generally higher than those detected by mass spec.
Line 75: please talk about fluctuations in testosterone in men.
Sex difference in physiology- sex steroids – this section will benefit from separating the section of steroid receptors from hormones. Also, there should be a parallel section that talks about pathophysiology associated with sex steroid biosynthesis pathways or perturbations in hormone levels.
Lines 194-95: what is the evidence and is that evidence really hold up? Just citing published papers is not helpful. There are caveats and loopholes in published data and the review would be more impactful if the authors really critiqued the data, rather than resummarizing.
In general, if the authors can organize the flow to talk about sex differences at baseline physiology/anatomy, followed by sex differences in pathophysiology/disease, mechanisms, if known, and treatments, if any and outcomes, if known. When discussing treatments, perhaps authors can point out if they were RCTs, placebo-controlled and whether sex ratio was skewed or not.
It would also be helpful to separate data from animal models or pre-clinical trials from human studies.
Overall, the review covers far too many topics to do justice to anyone. Perhaps the authors can consider splitting this review and cover addiction and pain in another review.
Lines 654-56: Did the authors really mean gender? Is there sufficient data. Please be careful about not using sex and gender interchangeably.
The immune and COVID aspect, if covered in this review should be separated from addiction.
Sex differences in vaccine responses is not really apt and has not been studied in detail. As mentioned before, the sheer number of topics covered in this review are numerous and thus the review appears diffuse and unfocused.
Reviewer 2 Report
The authors tackle a very interesting topic.
The gender difference with the different responses of the organism is today a known chapter but not too much.
I believe the review is extensively covered and useful for readers to improve population selection in clinical trials. However, I believe that the paper is too broad and the reader's attention is lost.
I suggest streamlining without losing the main focus.
Round 2
Reviewer 1 Report
The authors have addressed some concerns, the others are ignored or very superficially addressed. The review is very long and hard to navigate through.
The authors are referred to this publication to understand the differences between use of terms sex versus gender, and especially in clinical trials and factors to consider:
https://academic.oup.com/edrv/article/42/3/219/6159361?login=true
PMID: 33704446
Use of a table titles “The scenario described above is briefly summarized in Table xx” is odd and needs to be better stated.